# Positioning Error Analysis and Control of a Piezo-Driven 6-DOF Micro-Positioner

**DOI:** 10.3390/mi10080542

**Published:** 2019-08-17

**Authors:** Chao Lin, Shan Zheng, Pingyang Li, Zhonglei Shen, Shuang Wang

**Affiliations:** State Key Laboratory of Mechanical Transmission, Chongqing University, Chongqing 400030, China

**Keywords:** compliant mechanism, positioning error model, hysteresis nonlinearity, control compensation, micro-positioner

## Abstract

This paper presents a positioning error model and a control compensation scheme for a six-degree-of-freedom (6-DOF) micro-positioner based on a compliant mechanism and piezoelectric actuators (PZT). The positioning error model is established by means of the kinematic model of the compliant mechanism and complete differential coefficient theory, which includes the relationships between three typical errors (hysteresis, machining and measuring errors) and the total positioning error. The quantitative analysis of three errors is demonstrated through several experimental studies. Afterwards, an inverse Presiach model-based feedforward compensation of the hysteresis nonlinearity is employed by the control scheme, combined with a proportional-integral-derivative (PID) feedback controller for the compensation of machining and measuring errors. Moreover, a back propagation neural network PID (BP-PID) controller and a cerebellar model articulation controller neural network PID (CMAC-PID) controller are also adopted and compared to obtain optimal control. Taking the translational motion along the X axis as an example, the positioning errors are sharply reduced by the inverse hysteresis model with the maximum error of 12.76% and a root-mean-square error of 4.09%. In combination with the CMAC-PID controller, the errors are decreased to 0.63% and 0.23%, respectively. Hence, simulated and experimental results reveal that the proposed approach can improve the positioning accuracy of 6-DOF for the micro-positioner.

## 1. Introduction

Micro-positioners with piezoelectric actuation are widely utilized in plenty of important applications requiring ultrahigh-precision motion, such as in microelectronic mechanical systems (MEMS) [1], biological manipulation [2], atomic force microscopes [3], the lithographic machining of semiconductors [4], etc. Most micro-positioners rely on the elastic deformation of flexible hinges to create motion, but their contour-dependent deformation and motion behavior are currently difficult to predict [5], and have therefore been studied by many scholars. For example, an effective set of tractable equations for the rotational compliance of a simple monolithic flexure hinge was derived by the inverse conformal mapping of the circular approximating contour [6]. On the basis of this, a nonlinear parametric optimization of flexural hinge shapes was performed by Zelenika et al. [7], and this led to far better performances with respect to conventional circular notches. In addition, a flexible multibody model was developed in order to obtain the fixed and moving centrodes and the diameter of the inflection circle of the relative motion [8].

In recent years, micro-positioners have been implemented using a variety of different structures, ranging from single degree-of-freedom (DOF) to multiple DOF due to their wide and important application requirements. For example, a parallel 2-DOF flexible platform for XY nanopositioning was developed based on a novel multi-stage amplification mechanism [9]. Furthermore, a 3-DOF vertical micro-positioner was developed for optical instrument alignment [10], driven by three piezoelectric actuators (PZTs), guided by three rotationally symmetric hinges, and displacement-amplified by three bridge-type amplifiers. In addition, in order to meet the application requirements of spatial positioning, a 6-DOF series-parallel micro-positioner was proposed by Cai et al. [11]. It is worth noting that most micro-positioners can only be accurately positioned in the plane, and most of the stages that can achieve spatial 6-DOF positioning have small motion ranges and serious displacement coupling problems. The presented 6-DOF positioning stage has better decoupling performance and larger motion ranges [12]. In our previous work, the kinematics of the micro-positioner were explored, and this paper will continue to discuss its positioning error and control. 

Owing to the nanometer resolution, high stiffness, rapid response and large blocking force of PZT, micro-positioners driven by PZT have high positioning accuracy. Nevertheless, the serious hysteresis nonlinearity induced by PZT, the modeling and machining errors introduced by the compliant mechanism, and the vibration and detection problems attributable to the environment and sensor raise great challenges for the micro-positioner in achieving a precision position. Therefore, the availability of analysis methods for these errors is very necessary. For example, a geometric error model on account of the corresponding inverse and forward kinematic models was formulated, taking joint clearance-induced errors and structural parameter-induced errors into account [13]; a regularization method was proposed by Huang and Wang to identify the structure errors for a 3-PRS-XY (PRS: Prismatic-revolute-spherical) mechanism [14]; and the volumetric error model was established by means of homogeneous transformation matrices for a measurement-processing integrated machine tool [15].

Meanwhile, for compensating the geometric errors, a set of useful and ready-to-use indications was given in order to improve the kinematic accuracy by means of off-line kinematic calibration [16]. In addition, a common-path method was proposed to simultaneously measure the 6-DOF geometric motion errors and compensate the errors produced in the geometric measurements [17]. Several error measuring systems were also established for multi-degree-of-freedom positioning stages. A simultaneous measuring system for 5-DOF geometric error was shown to analyze and compensate the error crosstalk in the measuring procedure [18]. A capacitive position sensor for micro-positioning applications was generated to realize a high-precision X-Y linearity and rotational position accuracy [19]. Additionally, the hysteresis nonlinearity is generally deemed to be the main error source of PZT and can be described by building proper mathematical formulations to approximate the input-output behavior, such as the Preisach model [20], the Duhem model [21], the Maxwell model [22], the Dahl model [23], etc. The hysteresis nonlinearity was compensated by the open-loop feedforward control resorting to the inverse hysteresis models [24,25,26]. In addition, feedforward combined with close-loop feedback control was adopted for a preferable precision motion tracking [27,28,29]. Considering the nonlinearity as a disturbance or an uncertainty, several attempts have also been employed to apply the feedback control without modeling the inverse hysteresis, such as robust control [30,31], sliding model control [32,33] and robust adaptive control [34].

In previous research, error models were universally established for the geometric error of compliant mechanism, hardly incorporating other errors of micro-positioner into the proposed error models, especially for the complex mechanism with multiple DOF. Therefore, a positioning error model associated with the input variables induced by PZTs, structural parameters of compliant mechanism and measuring results impacted by the environment is proposed in this paper, aiming at building a relationship between the error sources and the positioning error. After that, the corresponding compensation strategies are proposed. By using a neural network algorithm for model identification, the forward and inverse Preisach models are established to respectively describe and compensate the hysteresis nonlinearity. Considering the machining error and measuring error as a disturbance or an uncertainty, a feedback control compensation strategy is more straightforward and effective than geometric calibration methods in the previous studies. In allusion to the feedback control, the proportional-integral-derivative (PID) control is still universally used nowadays in various fields owing to its simple control structure and ease of implementation and maintenance. Meanwhile, the back propagation neural network PID (BP-PID) control and cerebellar model articulation control neural network PID (CMAC-PID) control are also introduced and compared for feedback control in this paper to obtain optimal control.

This paper is organized as follows. The mechanical architecture and the positioning error model of the micro-positioner are described in Section 2. The inverse Preisach model, combined with PID, BP-PID and CMAC-PID controller are constructed for compensation of the positioning error in Section 3. The experimental verification for the positioning error model is conducted, and its control compensation in the 6-DOF is carried out by simulation and experiments in Section 4. Finally, Section 5 concludes this research. 

## 2. Positioning Error Analysis

### 2.1. Mechanism Description

The 6-DOF micro-positioner is illustrated in Figure 1a, and the overall dimensions measure 241 × 241 × 67 mm^3^. The micro-positioner is driven by PZTs and measured by capacitive displacement sensors to ensure fast response and high precision. It consists of a working platform and a compliant mechanism that is used to amplify and transmit the motion of PZTs via elastic deformations of the right angle flexure hinges. The compliant mechanism is composed of three parts, the top platform, the middle platform and the bottom platform, which are shown in Figure 1b–d, respectively. 

The parameter values of the micro-positioner are chosen based on the results of the optimization. To balance the static and dynamic performance of the micro-positioner, the parameter values are optimized by the MATLAB genetic algorithm optimization toolbox, which are listed in Table 1. The micro-positioner is designed as consistent geometric parameters in the same platform to eliminate the influence of nonlinear factors and improve the accuracy of the movement. 

To meet the design requirements of the large stroke, flexible hinge-based amplifiers must be adopted. At present, several major amplifiers are widely used, such as lever-type, bridge-type, rhombus-type and Scott-Russell mechanism. Among them, the bridge-type amplifier has the advantages of compact structure and large amplification ratio [35]; therefore, it is adopted by the presented stage. As shown in Figure 2a, when a voltage signal is input to the PZT placed in the X direction, the displacement and force generated by the PZT results in an amplified output displacement in the Y direction. 

For translation into the X/Y direction, it is controlled by the bottom platform. As shown in Figure 2b, when one of the two bridge-type amplifiers in the X direction is driven by PZT placed in it, a displacement $x_{out}$ in the X direction is generated, the middle platform and the top platform have the same displacement $x_{out}$ because of there being no constraint. As a result, the working platform receives a displacement $x_{out}$ in the X direction. Due to the symmetrical structure of the bottom platform, the working principle of the translation in the Y direction consistent with that in the X direction. 

The translation into the Z direction has a similar working principle, which is controlled by the middle platform. As shown in Figure 2c, when all four (or two symmetrical) bridge-type amplifiers in the middle platform are driven by PZT simultaneously, the top platform moves $z_{out}$ along the Z direction. At the same time, the working platform receives a displacement $z_{out}$ to achieve the function of Z-direction positioning.

With respect to rotation around the X/Y axis, this is also controlled by the middle platform. As shown in Figure 2d, when the bridge-type amplifier on the right is driven by the PZT, it moves up for Δ*Z*, and the platform has a rotation angle $\alpha_{out}$ around the X axis. Due to the symmetrical structure of the platform, the rotation around the Y axis is the same as the X axis. Therefore, the rotation around the X/Y axis can be controlled by means of controlling the displacement of the bridge-type amplifier in the middle platform.

As for the rotation around the Z axis, it is controlled by two bridge-type amplifiers in the top platform. As shown in Figure 2e, when the output displacement of both bridge-type amplifiers of the top platform is Δ*Y*, the platform has a rotation angle $\gamma_{out}$ around the Z axis.

### 2.2. Modeling Positioning Error

Positioning error is an important indicator for evaluating the performance of the micro-positioner. Since the micro-positioner consists of three stages, drive, positioning and measurement, the main errors are considered to be the driving error, machining error and measuring error, which work together to constitute the positioning error. The positioning error equation can be established as:(1)$$\delta \Delta_{\mathrm{out}}=\varepsilon_{p}+\varepsilon_{s}+\varepsilon_{c}$$

Referring to the pose description equation in [12], the motion of 6-DOF are expressed as implicit functions with geometric parameter and input displacement as variables: (2)$$\left\{ \begin{array}{l} f_{x}\left(x_{\mathrm{out}},l_{1},t_{1},l_{a1},l_{2},t_{2},l_{3},\alpha_{1},x_{\mathrm{pzt}}\right)=0\\ f_{y}\left(y_{\mathrm{out}},l_{1},t_{1},l_{a1},l_{2},t_{2},l_{3},\alpha_{1},y_{\mathrm{pzt}}\right)=0\\ f_{z}\left(z_{\mathrm{out}},l_{a2},t_{4},\alpha_{2},t_{4},z_{\mathrm{pzt}}\right)=0\\ f_{\alpha }\left(\alpha_{\mathrm{out}},l,l_{5},l_{a2},\alpha_{2},t_{4},\alpha_{\mathrm{pzt}}\right)=0\\ f_{\beta }\left(\beta_{\mathrm{out}},l,l_{5},l_{a2},\alpha_{2},t_{4},\beta_{\mathrm{pzt}}\right)=0\\ f_{\gamma }\left(\gamma_{\mathrm{out}},l_{6},t_{6},l_{a3},l_{7},t_{7},\alpha_{3},J,R,\gamma_{\mathrm{pzt}}\right)=0 \end{array}\right.$$

According to the differential relationship between the output displacement and the input displacement of the PZT, the driving error can be expressed as:(3)$$\varepsilon_{p}=-[\begin{array}{ccc} \frac{\partial f_{x}}{\partial x_{pzt}}\delta x_{pzt} & \frac{\partial f_{y}}{\partial y_{pzt}}\delta y_{pzt} & \frac{\partial f_{z}}{\partial z_{pzt}}\delta z_{pzt} \end{array}{\begin{array}{ccc} \frac{\partial f_{\alpha }}{\partial \alpha_{pzt}}\delta \alpha_{pzt} & \frac{\partial f_{\beta }}{\partial \beta_{pzt}}\delta \beta_{pzt} & \frac{\partial f_{\gamma }}{\partial \gamma_{pzt}}\delta \gamma_{pzt} \end{array}]}^{T}$$

The micro-positioner was monolithically fabricated by Wire cut Electrical Discharge Machining (WEDM) technology, and processing accuracy was 0.01–0.02 mm. The machining error of this study focuses on the geometric error caused by WEDM, which can be established as: (4)$$\varepsilon_{s}=-[\begin{array}{ccc} \sum_{i=1}^{7}\frac{\partial f_{x}}{\partial d_{xi}}\delta d_{xi} & \sum_{i=1}^{7}\frac{\partial f_{y}}{\partial d_{yi}}\delta d_{yi} & \sum_{i=1}^{3}\frac{\partial f_{z}}{\partial d_{zi}}\delta d_{zi} \end{array}{\begin{array}{ccc} \sum_{i=1}^{5}\frac{\partial f_{\alpha }}{\partial d_{\alpha i}}\delta d_{\alpha i} & \sum_{i=1}^{5}\frac{\partial f_{\beta }}{\partial d_{\beta i}}\delta d_{\beta i} & \sum_{i=1}^{8}\frac{\partial f_{\gamma }}{\partial d_{\gamma i}}\delta d_{\gamma i} \end{array}]}^{T}$$
where,
(5)$$\left\{ \begin{array}{l} {\left[d_{xi}\right]}_{\times 7}={\left[d_{yi}\right]}_{\times 7}=\left[l_{1},t_{1},l_{a1},l_{2},t_{2},l_{3},\alpha_{1}\right]\\ {\left[d_{\gamma i}\right]}_{\times 6}=\left[l_{6},t_{6},l_{a3},l_{7},t_{7},\alpha_{3},J,R\right]\\ {\left[d_{zi}\right]}_{\times 3}=\left[l_{a2},\alpha_{2},t_{4}\right]\\ {\left[d_{\alpha i}\right]}_{\times 5}={\left[d_{\beta i}\right]}_{\times 5}=\left[l,l_{5},l_{a2},\alpha_{2},t_{4}\right] \end{array}\right.$$
$l_{k}(k=1,2,3,5,6,7,a_{1},a_{2},a_{3})$, l, J, R, $\alpha_{j}\left(j=1,2,3\right)$ and $t_{s}(s=1,2,4,6,7)$ are all the geometric parameters of micro-positioner as shown in Figure 1.

The measuring error is connected with the resolution of capacitive displacement sensors and the environmental disturbance. The resolution of capacitive displacement sensors is less than 0.1 μm, and the detection precision is up to 0.02%, which is in accordance with the experimental requirements. Hence, the environmental disturbance is regarded as the main source of measurement error, and this is reflected in the noise signal of experimental output in the working platform. The positioning error of measurement can be acquired directly by experiments and written as:(6)$$\varepsilon_{c}={\left[\begin{array}{cccccc} \delta x_{cp} & \delta y_{cp} & \delta z_{cp} & \delta \alpha_{cp} & \delta \beta_{cp} & \delta \gamma_{cp} \end{array}\right]}^{T}$$

The micro-positioner is driven by PZT; however, the positioning performance is degraded severely because of the hysteresis nonlinearity of PZT. Therefore, the hysteresis nonlinearity is regarded as the main source of driving error. Driving error is further resolved as:(7)$$\delta w_{pzt}=\left|w_{pzt}\left(t\right)-\varsigma \cdot u_{pzt}^{w}\right|=\left|H\left(u_{pzt}^{w}\right)-\varsigma \cdot u_{pzt}^{w}\right|$$

To further calculate the hysteresis nonlinearity, the Preisach model is proposed as:(8)$$w_{pzt}\left(t\right)=H\left(u_{pzt}^{w}\right)=\iint_{\alpha \geq \beta }\mu \left(\hat{\alpha },\hat{\beta }\right){\hat{\gamma }}_{\alpha \beta }\left[u_{pzt}^{w}\left(t\right)\right]d\hat{\alpha }d\hat{\beta }$$

There exists a one-to-one correspondence between the hysteresis operator ${\hat{\gamma }}_{\alpha \beta }\left[u_{pzt}^{w}\left(t\right)\right]$ with an output of +1 or −1 and the switching value pair $\left(\hat{\alpha },\hat{\beta }\right)$. This model may be reasonably approximated by a finite superposition of different rectangular operators.
(9)$$w_{pzt}\left(t\right)\approx \sum_{i=1}^{N}\sum_{j=1}^{N}\mu \left({\hat{\alpha }}_{i},{\hat{\beta }}_{j}\right){\hat{\gamma }}_{\alpha_{i}\beta_{j}}\left[u_{pzt}^{w}\left(t\right)\right],{\hat{\alpha }}_{i}={\hat{\beta }}_{i}=\frac{N-i}{N-1}{\hat{\alpha }}_{1}$$
where ${\hat{\alpha }}_{1}$ indicates the input voltage at which positive saturation of the actual hysteresis loop is achieved. Meanwhile, ${\hat{\alpha }}_{1}$ is divided into N, resulting in N(N+1)/2 switching value pairs $\left(\hat{\alpha },\hat{\beta }\right)$.

In order to solve this model, the Preisach function, which defines the changing amount of output displacement when the input voltage drops from $\hat{\alpha }$ to $\hat{\beta }$, is promoted as: (10)$$P_{pzt}^{w}\left({\hat{\alpha }}^{\prime },{\hat{\beta }}^{\prime }\right)=w_{{\hat{\alpha }}^{\prime }}-w_{{\hat{\alpha }}^{\prime }{\hat{\beta }}^{\prime }}$$

Considering the solution process of the Preisach function in [36], the relationship between displacement and voltage is derived on the basis of two states: the voltage rising process and the voltage falling process. The general mathematical form of the Preisach model was employed as follows: (11)$$\left\{ \begin{array}{l} w_{pzt}\left(t\right)=\sum_{k=1}^{n}\left[P_{pzt}^{w}\left({\hat{\alpha }}_{k}^{\prime },{\hat{\beta }}_{k-1}^{\prime }\right)-P_{pzt}^{w}\left({\hat{\alpha }}_{k}^{\prime },{\hat{\beta }}_{k}^{\prime }\right)\right]+P_{pzt}^{w}\left(u_{pzt}^{w}\left(t\right),{\hat{\beta }}_{n}^{\prime }\right)\\ w_{pzt}\left(t\right)=\sum_{k=1}^{n-1}\left[P_{pzt}^{w}\left({\hat{\alpha }}_{k}^{\prime },{\hat{\beta }}_{k-1}^{\prime }\right)-P_{pzt}^{w}\left({\hat{\alpha }}_{k}^{\prime },{\hat{\beta }}_{k}^{\prime }\right)\right]+\left[P_{pzt}^{w}\left({\hat{\alpha }}_{n}^{\prime },{\hat{\beta }}_{n-1}^{\prime }\right)-P_{pzt}^{w}\left(\alpha_{n}^{\prime },u_{pzt}^{w}\left(t\right)\right)\right] \end{array}\right.$$
where, it should be noted that the first one is the description of the relationship between displacement and voltage when voltage is rising, and the second one is that when voltage is falling.

## 3. Positioning Error Compensation Design

### 3.1. System Identification

Owing to the structural symmetry and motion decoupling of the micro-positioner, the 6-DOF motion can be regarded as equivalent to six independent single-input single-output (SISO) motions. Additionally, the translational motion or rotational motion along the X axis and the Y axis is in full accord. For the sake of brevity, only the treatment of X-axis translational motion is presented in this section, and the motions of the other axes are equally available. As depicted in Figure 3, the compliant mechanism in a single axis is simplified as a mass-spring-damper system with the equivalent mass, stiffness and damping coefficient of $m_{w}$, $k_{w}$ and c; PZT, which makes use of the inverse piezoelectric effect to convert the driving voltage of $u_{pzt}^{w}$ into a driving force of $F_{pzt}^{w}$ in the form of electric charge q, is generally considered to be a capacitor with an equivalent capacitance to $C_{C}$. Meanwhile, the driving circuit of the PZT controller can be simplified as an amplifier with an amplification ratio of $k_{C}$ and an equivalent resistance of $R_{C}$.

Based on the electromechanical coupling model shown in Figure 3 and Kirchhoff’s law, the voltage relationship of the PZT is established as: (12)$$R_{c}C_{c}\frac{du_{pzt}^{w}}{dt}+u_{pzt}^{w}=u_{\mathrm{in}}^{w}$$

Since the PZT is composed of many ceramic sheets, the relationship between the input voltage at both ends and the no-load output displacement can be computed as:(13)$$w_{pzt0}=n\cdot d\cdot u_{pzt}^{w}$$

Due to the mechanical structure of the PZT external connection with certain stiffness, some of the output displacement will be lost, and the output displacement of the PZT can be expressed as:(14)$$w_{pzt}=\frac{k_{pzt}}{k_{pzt}+k_{in}^{v}}\cdot w_{pzt0}$$

Combining Equations (12)–(14), the transfer function of PZT can be derived as:(15)$$G_{pzt}^{w}\left(s\right)=\frac{w_{pzt}\left(s\right)}{u_{c}^{w}\left(s\right)}=\frac{ndk_{pzt}/\left(k_{pzt}+k_{in}^{v}\right)}{\left(R_{c}C_{c}s+1\right)}$$

Considering the output displacement loss of PZT when the mechanical structure of the external connection of it has rigidity, the linear dynamic model can be established as follows:(16)$$\left\{ \begin{array}{l} m_{w}{\ddot{w}}_{out}+c_{w}{\dot{w}}_{out}+k_{w}w_{out}=F_{in}^{w}\\ F_{in}^{w}=k_{in}^{v}\cdot w_{pzt} \end{array}\right.$$

The ideal transfer function of the platform can be achieved as:(17)$$G_{stage}^{w}\left(s\right)=\frac{w_{out}\left(s\right)}{w_{pzt}\left(s\right)}=\frac{k_{in}^{v}}{m_{w}s^{2}+cs+k_{w}}$$

Combining Equations (15) and (17), the transfer function of the entire system can be derived as: (18)$$G_{w}\left(s\right)=G_{pzt}^{w}\left(s\right)\cdot G_{stage}^{w}\left(s\right)=\frac{w_{out}\left(s\right)}{u_{in}^{w}\left(s\right)}=\frac{ndh_{c}k_{pzt}k_{in}^{v}/\left(k_{pzt}+k_{in}^{v}\right)}{\left(R_{c}C_{c}s+1\right)\left(m_{w}s^{2}+cs+k_{w}\right)}$$

### 3.2. Feedforward Compensation

For the sake of compensating the hysteresis nonlinearity of PZT and eliminating the driving error of the micro-positioner, a feedforward compensation is proposed. For a given position trajectory, the expected input voltage can be calculated by the inverse Preisach model from the Preisach model in Equation (11).

(19)$$\left\{ \begin{array}{l} u_{pzt}^{w}\left(t\right)=P_{pzt}^{w}{}^{-1}(w_{pzt}^{d}\left(t\right)-\sum_{k=1}^{n}[P_{pzt}^{w}\left({\hat{\alpha }}_{k}^{\prime },{\hat{\beta }}_{k-1}^{\prime }\right)-P_{pzt}^{w}\left({\hat{\alpha }}_{k}^{\prime },{\hat{\beta }}_{k}^{\prime }\right)],{\hat{\beta }}_{n}^{\prime })\\ u_{pzt}^{w}\left(t\right)=P_{pzt}^{w}{}^{-1}({\hat{\alpha }}_{k}^{\prime },\sum_{k=1}^{n-1}[P_{pzt}^{w}\left({\hat{\alpha }}_{k}^{\prime },{\hat{\beta }}_{k-1}^{\prime }\right)-P_{pzt}^{w}\left({\hat{\alpha }}_{k}^{\prime },{\hat{\beta }}_{k}^{\prime }\right)]+\left[P_{pzt}^{w}\left({\hat{\alpha }}_{n}^{\prime },{\hat{\beta }}_{n-1}^{\prime }\right)-w_{pzt}^{d}\left(t\right)\right]) \end{array}\right.$$

The inverse Preisach model is described as: (20)$$u_{pzt}^{w}=H^{-1}\left(w_{pzt}^{d}\right)$$
where $H\left(u_{pzt}^{w}\right)$ and $H^{-1}\left(w_{pzt}^{d}\right)$ can be regarded as the mapping and inverse mapping, and the hysteresis nonlinearity of PZT will be offset by the inverse model based on Preisach model: (21)$$w_{pzt}=H\left(u_{pzt}^{w}\right)=H\left[H^{-1}\left(w_{pzt}^{d}\right)\right]\approx w_{pzt}^{d}$$

Therefore, the inverse model is in series with the hysteresis model as a filter, so as to obtain a control signal that can achieve accurate tracking for a given displacement of PZT.

### 3.3. Feedforward Plus Feedback Compensation

The driving error was predominantly eliminated by the feedforward compensation in Equation (21), but the machining error and measurement error described in Section 2 still exist; therefore, an additional feedback controller should be adopted. The concept for this is illustrated in Figure 4, where the PID, BP-PID and CMAC-PID controllers are employed in turn as the control algorithm for determining the optimal controller. 

On basis of the feedforward compensation, the positioning error of micro-positioner is rewritten as: (22)$$\delta w_{out}=w_{pzt}^{d}\cdot \lambda -w_{out}$$

Which meets the following conditions: (23)$$w_{pzt}^{d}\approx w_{pzt},\varepsilon_{p}\approx 0$$

The PID algorithm is described as: (24)$$w_{pzt}^{p}=K_{p}\left[\delta w_{out}\left(t\right)+\frac{1}{T_{i}}\int_{0}^{t}\delta w_{out}\left(\tau \right)d\tau +T_{d}\frac{\delta w_{out}\left(t\right)}{dt}\right]$$

The BP neural network is introduced in series with the PID controller to adjust parameters. It is a three-layer network comprising an input layer, a hidden layer and an output layer. The output layer is a combination of the hidden layer and itself, and both layers consist of an array of functions. The algorithm permits a more effective weight updating procedure for infinite approximation of a smooth function f(θ), and the output vector of the function is the three control parameters of PID. Thus, the BP neural network can be expressed as: (25)$$f\left(\theta \right)=g\left\{W^{\left(2\right)}\cdot \left[\phi \left(W^{\left(1\right)}\cdot \theta \right)-\rho^{\left(1\right)}\right]\right\}-\rho^{\left(2\right)}$$

The input matrix θ and the output matrix f(θ) are presented as: (26)$$\theta ={\left[\begin{array}{ccc} \delta {\dot{w}}_{out} & \delta w_{out} & \delta {\ddot{w}}_{out} \end{array}\right]}^{T},f\left(\theta \right)={\left[\begin{array}{ccc} K_{p} & T_{i} & T_{d} \end{array}\right]}^{T}$$

The weight matrix $W^{\left(1\right)}$ and threshold matrix $\rho^{\left(1\right)}$ contain the weights $\omega_{11}^{h}\cdots \omega_{33}^{h}$ and the ideal thresholds $\rho_{1}^{h}\cdots \rho_{3}^{h}$ of the hidden layer; and the weight matrix $W^{\left(2\right)}$ and threshold matrix $\rho^{\left(2\right)}$ contain $\omega_{11}^{o}\cdots \omega_{33}^{o}$ and $\rho_{1}^{o}\cdots \rho_{3}^{o}$ of the output layer similarly. They are shown as: (27)$$\rho^{\left(1\right)}={\left[\begin{array}{ccc} \rho_{1}^{h} & \rho_{2}^{h} & \rho_{3}^{h} \end{array}\right]}^{T},\rho^{\left(2\right)}={\left[\begin{array}{ccc} \rho_{1}^{o} & \rho_{2}^{o} & \rho_{3}^{o} \end{array}\right]}^{T}$$

(28)$$W^{\left(1\right)}=\left[\begin{array}{ccc} \omega_{11}^{h} & \omega_{12}^{h} & \omega_{13}^{h}\\ \omega_{21}^{h} & \omega_{22}^{h} & \omega_{23}^{h}\\ \omega_{31}^{h} & \omega_{32}^{h} & \omega_{33}^{h} \end{array}\right],W^{\left(2\right)}=\left[\begin{array}{ccc} \omega_{11}^{o} & \omega_{12}^{o} & \omega_{13}^{o}\\ \omega_{21}^{o} & \omega_{22}^{o} & \omega_{23}^{o}\\ \omega_{31}^{o} & \omega_{32}^{o} & \omega_{33}^{o} \end{array}\right]$$

Moreover, the activation functions of the hidden layer and output layer are expressed as:(29)$$\phi \left(w\right)=\frac{e^{w}-e^{-w}}{e^{w}+e^{-w}},g\left(w\right)=\frac{e^{w}}{e^{w}+e^{-w}}$$

Substituting Equation (25) into Equation (24), the output of the BP-PID controller can be readily acquired. Meanwhile, in order to continuously optimize three parameters of the PID controller, the performance indicator function is taken as: (30)$$E=\frac{1}{2}{\left(w_{pzt}^{d}\cdot \lambda -w_{out}\right)}^{2}$$

Then, the function is employed by means of the gradient descent method (GDM) to update the procedure and hunt for the most effective weights and thresholds. 

Due to the uncertainty of the weight initialization and the number of hidden layer nodes, the BP-PID controller affects the convergence speed, complexity and results of the computational process. Therefore, the CMAC neural network algorithm is added in parallel to the existing PID controller. This algorithm uses the output of the PID controller $w_{pzt}^{p}$ to train the CMAC neural network $w_{pzt}^{n}$, which minimizes the difference between the total control output $w_{pzt}$ and the CMAC control contribution. Hence, the learning algorithm is obtained as: (31)$$w_{pzt}^{n}=\sum_{i=1}^{c}\omega_{i}a_{i}$$

(32)$$w_{pzt}=w_{pzt}^{n}+w_{pzt}^{p}$$

Similarly, the performance indicator function is as follows:(33)$$E=\frac{1}{2}{\left(w_{pzt}-w_{pzt}^{n}\right)}^{2}\cdot \frac{a_{i}}{\chi }$$

Then the weight $\omega_{i}$ can be updated constantly to optimum. In addition, compared with BP-PID controller, this algorithm modifies fewer weights and converges faster, which has nonlinear approximation ability, adaptability and easy implementation.

## 4. Simulation, Experiment and Discussion

On basis of the above research, the quantitative relationship between the positioning error and the three errors, the driving error, machining error and measuring error, will be further analyzed and verified by experiments in this section. Meanwhile, the positioning error is embodied in the open-loop experiment of micro-positioner, and can be well compensated and verified by the simulations and experiments.

### 4.1. Experimental Setup

The experimental setup of the micro-positioner is graphically shown in Figure 5a, where the fixed holes of micro-positioner is mounted on an optical table. The micro-positioner is fabricated from a piece of material (the spring steel 60Si_2_Mn, SHANGHAI DIHUA METAL MATERIAL, Inc., Shanghai, China) by Wire cut Electric Discharge Machining (WEDM) technology. PZTs (model Pst-40VS15 produced by COREMORROW, Inc., Harbin, China) are adopted to drive the micro-positioner, and PZTs are actuated with a voltage of 0–120 V through the PZT controller (XE-500/501 series modular controller, AYP Nano Solutions, Inc., Anaheim, CA, USA), deriving the maximum displacement of 40 μm. The output displacement of micro-positioner is measured by capacitive displacement sensors (model CS5, from MICRO-EPSILION, Inc., Raleigh, NC, USA) with a 100 nm resolution and a 5 mm measurement range, and the signals of the sensors are sent to the host computer through the sensor controller (model RS6500, MICRO-EPSILION, Inc., Raleigh, NC, USA). The hardware connection between the instruments is illustrated in Figure 5b.

### 4.2. Experiments of Positioning Error

Because the driving error is mainly derived from the hysteresis nonlinearity of PZT, the experiment for hysteresis nonlinearity is performed to obtain the hysteresis curve and verify how the built model in Section 2 fits the actual hysteresis. The sample data points are established by dividing the input voltage into N sets of voltage groups and experimentally measuring the corresponding output displacements. Then the sample data points are brought into the artificial neural networks (ANN) toolbox in Matlab software (The MathWorks, Natick, MA, USA) to identify the parameters of Preisach model. Finally, the simulated results for just the continuous voltage-displacement curve is obtained. The comparison between the experimental results and the simulated results is shown in Figure 6.

It can be observed that the width of the hysteresis is wider for a higher input voltage amplitude, differing enormously from the ideal linear output of PZT. To evaluate the hysteresis error, the maximum hysteresis error is defined as the ratio of the maximum distance at the same voltage value on the hysteresis loop to the maximum displacement value on the hysteresis loop, which is 12.83% and can be expressed as:(34)$$e_{loop}\left(\%\right)=\frac{\max\left|x_{up}\left(u_{0}\right)-x_{down}\left(u_{0}\right)\right|}{\max\left(x\right)}\times 100\%$$
where x is the desired displacement; $x_{up}\left(u_{0}\right)$ and $x_{down}\left(u_{0}\right)$ are the predicted output displacements of the up and down processes on the hysteresis loop at which the voltage is $u_{0}$, respectively; $u_{0}$ is any voltage on the hysteresis loop.

Meanwhile, it is clear that the proposed model can well describe the hysteresis nonlinearity of the micro-positioner. The maximum positioning error between them is $e_{max}=1.43\%$ and the root mean square error is $e_{rms}=0.28\%$, they are defined as:(35)$$e_{\max}\left(\%\right)=\max\left|\frac{x_{d}-x}{\max\left(x_{d}\right)-\min\left(x_{d}\right)}\right|\times 100\%$$
(36)$$e_{rms}\left(\%\right)=\left(\frac{\sqrt{\frac{1}{N}\sum_{i=1}^{N}e_{i}^{2}}}{\max\left(x_{d}\right)-\min\left(x_{d}\right)}\right)\times 100\%$$
where $x_{d}$ is the measured displacements; N is the number of data; and $e=x_{d}-x$ is the tracking error.

On the basis of provisionally not considering the driving and measuring errors, the machining error of the micro-positioner that is the relationship between the positioning error and the geometric parameters, can be obtained by bringing the measuring and designed data of the geometric parameters into Equation (4).

For moving along X/Y axis in Figure 7a, the geometric parameters of the bottom platform ($t_{1}$,$t_{2}$,$l_{1}$,$l_{a1}$) have a great influence on the positioning error; in order to move along the Z axis in Figure 7b, the geometric parameter of the middle platform ($t_{4}$) gradually becomes larger, while the parameter of $l_{a2}$ sharply rises and then falls rapidly to be gentle, revealing that the parameter of $t_{4}$ affects the positioning error most. Similarly, for rotating along X/Y axis and Z axis in Figure 7c,d, the geometric parameters of l, $t_{6}$, $t_{7}$ and $l_{6}$ have the most severe impact, while the parameters of $l_{5}$ and $t_{4}$ are so small that can be neglected. Although the influence of geometric parameters on positioning error can be reduced by optimizing structural design, there still exists a certain machining error that needs to compensate.

Owing to the high resolution and detection accuracy of the capacitive displacement sensor, the environmental disturbance is considered to be the main source of the measuring error. On account of the analogous property in 6-DOF for the measuring error, taking the translational motion along X/Y axis as an example, the environmental disturbance that can be reflected by the output of the micro-positioner in the stationary state is detected in Figure 8. It is demonstrated that the signal is generally stable in the range of 0.05–0.15 μm, with an average value of 0.103 μm and a standard deviation of 3.18%. The phenomenon indicates that the measuring error is tiny and concentrated, which can be roughly assumed by the average value.

### 4.3. Simulated Testing of Error Compensation

For the positioning error of the micro-positioner, the inverse Preisach model is used as the feedforward to compensate the driving error, and the PID, BP-PID, CMAC-PID controllers are used as feedback to compensate the machining and measuring errors, etc. For the sake of simplicity, the simulated testing is carried out with the translational motion along X/Y axis, and other degrees of freedom are equally available.

Taking the hysteresis loop with the input voltage amplitude of 120 V as an example, the inverse Preisach model is identified through back propagation neural network algorithm and Matlab to compensate the hysteresis, as illustrated in Figure 9. It can be found that the shapes of the inverse hysteresis loop and the hysteresis loop in Figure 9a,b are roughly symmetric distribution of y = x axis. Furthermore, with the inverse hysteresis loop superimposed on the hysteresis loop, the input–output relationship is converted from the nonlinearity of voltage–displacement to the linearity of displacement–displacement in Figure 9c. In addition, the rising and falling curves basically coincide with a straight line, with a maximum gap of 1.23 μm and an average error of 0.64 μm. The maximum hysteresis error decreased to a low level of 3.02% in comparison to the initial error of 12.83%. In addition, the compensated output is compared to the ideal input with $e_{max}$ of 1.84% and $e_{rms}$ of 0.65% in Figure 9d, satisfying the positioning accuracy of 5%. Hence, the inverse Preisach model can preferably eliminate the hysteresis nonlinearity, and greatly reduce the driving error.

Based on the feedforward compensation above and the controller design in Section 3, the closed-loop response of the micro-positioner can be acquired by taking the unit step curve as the input signal, as depicted in Figure 10. Quantitatively tabulating the results of the designed controllers in Table 2, it is observed that the PID and BP-PID controllers can absolutely eliminate the overshoot, respectively shortening the settling time by 0.16 s and 0.38 s, and promoting the settling value by 0.059 μm and 0.113 μm compared with the desired value of 6.613. Although the CMAC-PID controller gives a moderate response speed with about 1.39% overshoot, the response can reach stability with the quickest transient response and obtain the final value closest to the ideal. Therefore, CMAC-PID controller is chosen to the optimal, thanks to the best approving control effect comparatively.

### 4.4. Positioning Error Compensation Experiment

For verifying the simulated results above, a sinusoidal motion trajectory with 0.5 Hz frequency and 2 μm peak-to-peak amplitude is tracked with the inverse hysteresis compensation (HC) and the three types of controllers, and the results are compared in Figure 11a,c. From the magnitude of the tracking errors as depicted in Figure 11b,d and described in Table 3, it is obviously implied that the inverse hysteresis compensation reduces the positioning error sharply by the maximum error of 12.76% and the root-mean-square error of 4.09%. Moreover, CMAC-PID controller is superior to both PID and BP-PID controllers in terms of $e_{max}$ and $e_{rms}$. Compared with PID controller that generates $e_{max}$ of 2.91% and $e_{rms}$ of 1.38% respectively, BP-PID controller can merely suppress the errors around 1.07% and 0.62%. Nevertheless CMAC-PID controller substantially decreases the errors to 0.63% and 0.23% in contrast, verifying the optimality.

With the inverse hysteresis compensation and the optimal controller of CMAC-PID combined, the positioning error compensation of the other four degrees of freedom is conducted as illustrated in Figure 12. It can be seen from Table 4 that $e_{max}$ and $e_{rms}$ values of CMAC-PID controller based on the feedforward compensation for the translational motion along Z axis are reduced by 9.85% and 4.55%, similar to the characteristics of that along X/Y axis. Furthermore, although $e_{max}$ and $e_{rms}$ values of the open-loop experiment in the rotational motion of X/Y axis are very large to 30.77% and 16.10%, they are drastically reduced by up to 23.8% and 13.04% via compensation. Similarly, the positioning errors in terms of $e_{max}$ and $e_{rms}$ are sharply decreased by up to 35.15% and 18.02% in the rotational motion of Z axis.

Furthermore, the open-loop positioning errors of the rotations are significantly 3–5 times higher than that of the translations, revealing that the experiments of the translations work better and preferably reflect the characteristics. Meanwhile, in spite that the error compensation effects of the translations are still superior to those of the rotations, the positioning errors of $e_{max}$ and $e_{rms}$ in the rotations are distinctly suppressed to within 20% and 10%, prominently improving the positioning accuracy.

## 5. Conclusions

In this paper, a positioning error model of the micro-positioner is established to analyze the total positioning error in 6-DOF concretely, and the corresponding compensation scheme is proposed. It can be demonstrated from experiments that the positioning error model is an efficient method to reflect the relationship between three partial errors and the total positioning error in 6-DOF quantitatively. Furthermore, the Preisach model is capable of describing accurately the driving error of non-symmetric piezoelectric hysteresis. Simulated and experimental results imply that the combined inverse Presiach model and PID feedback controller can compensate the nonlinearity of system better than either of the stand-alone control schemes. Moreover, the CMAC-PID controller can improve the tracking performance with the maximum error and root-mean-square error in comparison to both the PID and BP-PID controllers. Meanwhile, the proposed methodology can be easily extended to other types of micro-positioners actuated by PZT as well.

## Figures and Tables

**Figure 1 micromachines-10-00542-f001:**
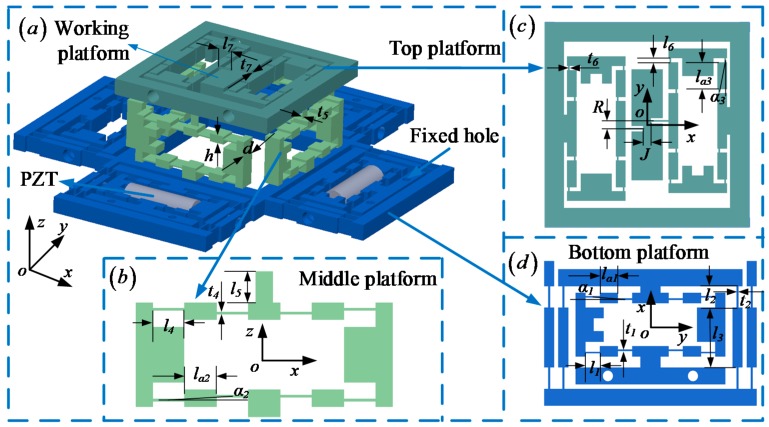
Schematic diagram of the six-degree-of-freedom (6-DOF) micro-positioner. (**a**) The overall structure of the micro-positioner, (**b**) the structure of the middle platfoem, (**c**) the structure of the top platfoem, (**d**) the structure of the bottom platfoem.

**Figure 2 micromachines-10-00542-f002:**
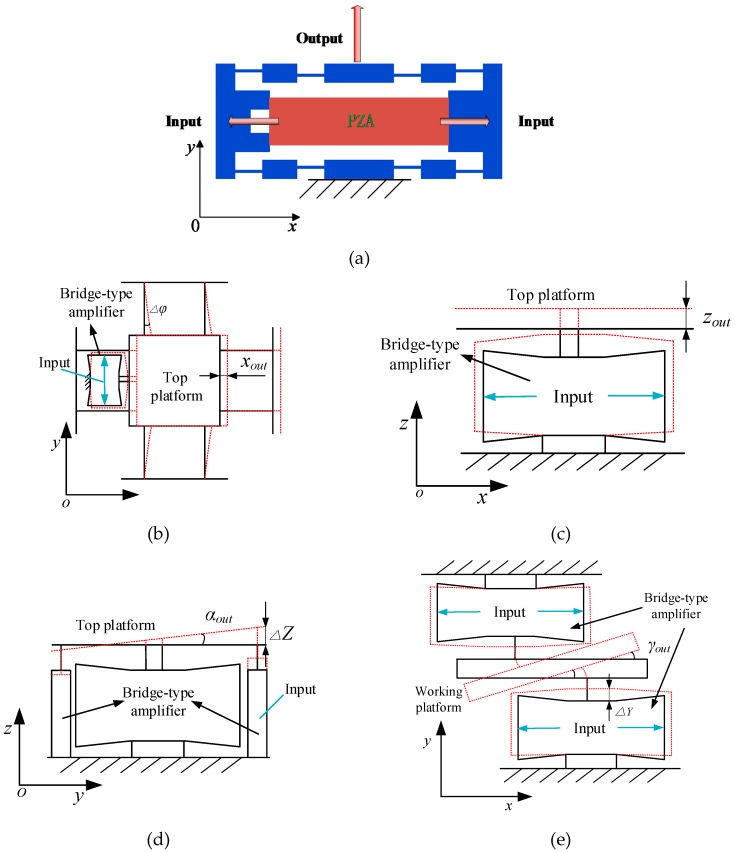
Schematic diagram of the stage: (**a**) schematic of the bridge-type amplifier, (**b**) schematic of translation in the X direction, (**c**) schematic of translation in the Z direction, (**d**) schematic of rotation around the X axis and (**e**) schematic of rotation around the Z axis.

**Figure 3 micromachines-10-00542-f003:**
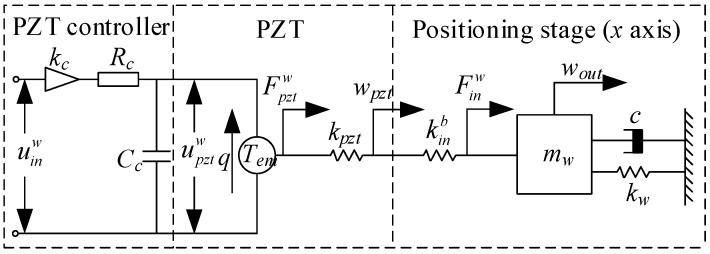
Simplified electromechanical coupling model.

**Figure 4 micromachines-10-00542-f004:**
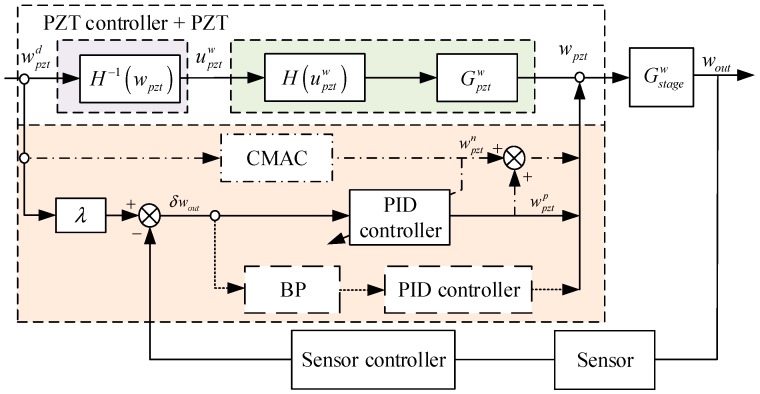
Block diagram of the feedforward plus feedback compensation.

**Figure 5 micromachines-10-00542-f005:**
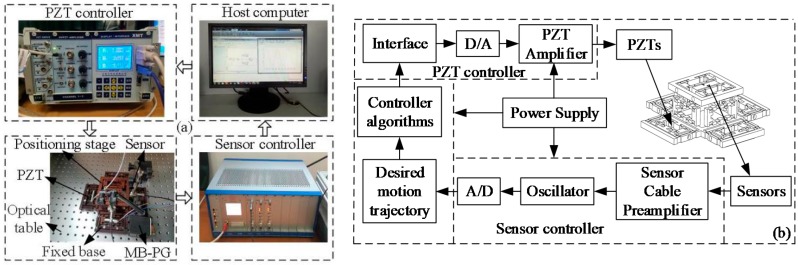
Schematic diagram of the 6-DOF micro-positioner: (**a**) experimental setup, (**b**) hardware connection.

**Figure 6 micromachines-10-00542-f006:**
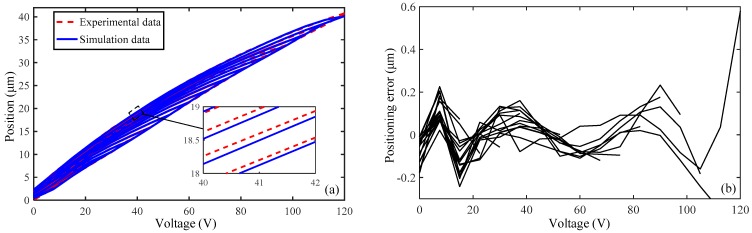
Comparison of the hysteresis loop between the experimental results and the simulated results: (**a**) hysteresis loop, (**b**) hysteresis error.

**Figure 7 micromachines-10-00542-f007:**
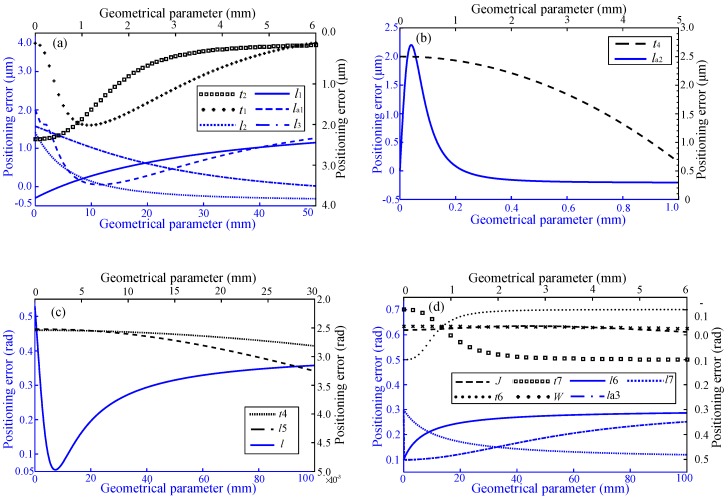
Machining error results for (**a**) moving along X/Y axis, (**b**) moving along Z axis, (**c**) rotating along X/Y axis, and (**d**) rotating along Z axis.

**Figure 8 micromachines-10-00542-f008:**
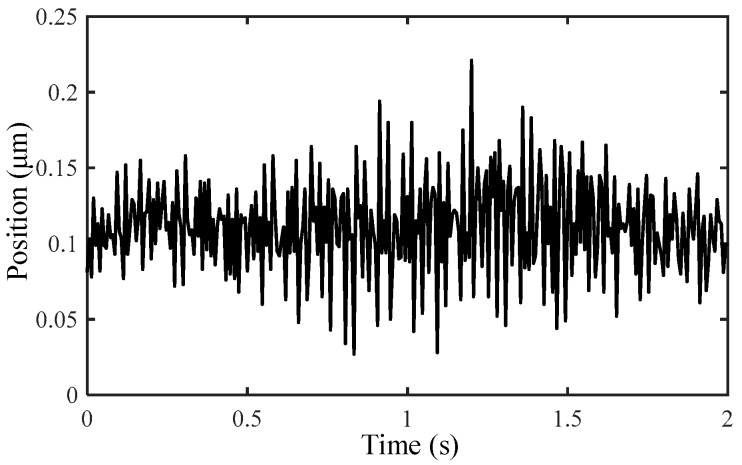
Environmental disturbance of the micro-positioner.

**Figure 9 micromachines-10-00542-f009:**
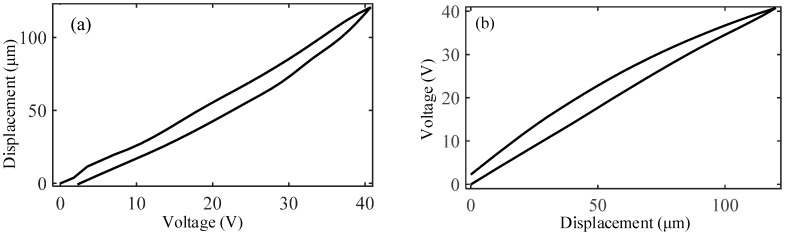
Schematic diagram of the inverse Preisach model compensation: (**a**) the hysteresis loop, (**b**) the inverse hysteresis loop, (**c**) the compensated output, and (**d**) the compensated output error.

**Figure 10 micromachines-10-00542-f010:**
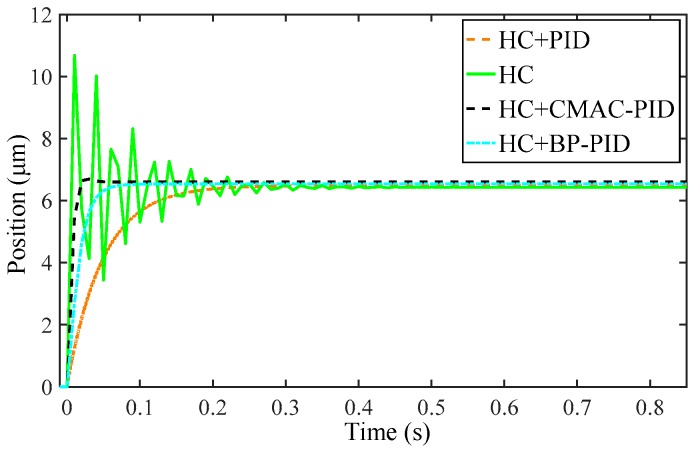
Close-loop response simulation of the micro-positioner.

**Figure 11 micromachines-10-00542-f011:**
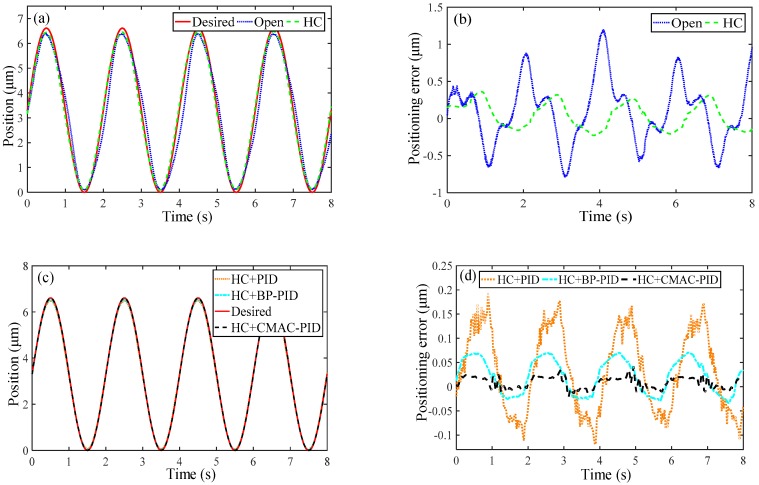
Experimental motion tracking results of micro-positioner for the translational motion along X/Y axis: (**a**,**b**) feedforward compensation, (**c**,**d**) feedback compensation.

**Figure 12 micromachines-10-00542-f012:**
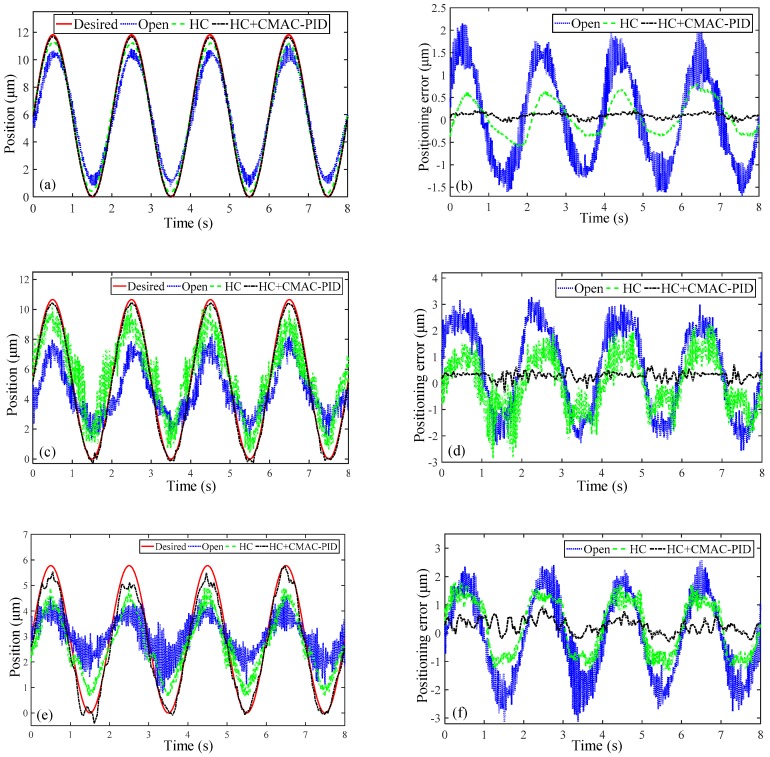
Experimental tracking results of (**a**,**b**) the translation along Z axis, (**c**,**d**) the rotation along X/Y axis, (**e**,**f**) the rotation along Z axis.

**Table 1 micromachines-10-00542-t001:** The parameter values of the micro-positioner.

Top Platform		Middle Platform		Bottom Platform	
*l_6_* (mm)	2	*l_4_* (mm)	9	*l_1_* (mm)	7
*t_6_* (mm)	0.8	*t_4_* (mm)	0.8	*t_1_* (mm)	0.8
*l_a3_* (mm)	14	*l_a2_* (mm)	9	*l_a1_* (mm)	9
*α_3_* (rad)	0.185	*α_2_* (rad)	0.061	*α_1_* (rad)	0.068
*l_7_* (mm)	2	*l_5_* (mm)	9	*l_2_* (mm)	10
*t_7_* (mm)	0.8	*t_5_* (mm)	0.8	*t_2_* (mm)	1
*R* (mm)	4	*h* (mm)	5	*l_3_* (mm)	28
*J* (mm)	4	*d* (mm)	10	_	

**Table 2 micromachines-10-00542-t002:** Controller performance for compensation simulation.

Performance	HC	HC+PID	HC+BP-PID	HC+CMAC-PID
Settling time(sec)	0.84	0.68	0.46	0.11
Overshoot (%)	66.12	0	0	1.39
Settling value(μm)	6.429	6.488	6.542	6.601
Positioning error (%)	2.78	1.89	1.07	0.18

**Table 3 micromachines-10-00542-t003:** Experimental controller performance for the translational motion along X/Y axis.

Performance	OPEN	HC	HC+PID	HC+BP-PID	HC+CMAC-PID
$e_{\max}$(%)	18.22	5.46	2.91	1.07	0.63
$e_{rms}$(%)	6.70	2.61	1.38	0.62	0.23

**Table 4 micromachines-10-00542-t004:** Experimental controller performance for the remaining degrees of freedom.

Freedom	Translation Along Z Axis			Rotation Around X/Y Axis			Rotation Around Z Axis		
Performance	OPEN	HC	HC+CMAC-PID	OPEN	HC	HC+CMAC-PID	OPEN	HC	HC+CMAC-PID
*e_max_* (%)	11.62	6.79	1.77	30.77	22.16	6.97	54.17	31.53	19.02
*e_rms_* (%)	50	3.30	0.95	16.10	10.33	3.06	25.34	16.82	7.32

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
