# Peer review of "Positioning Error Analysis and Control of a Piezo-Driven 6-DOF Micro-Positioner"

_micromachines, 2019, doi:10.3390/mi10080542_

Round 1

Reviewer 1 Report

The submitted paper presents the analysis and control of a piezo-driven 6 DOF micro-positioning system based on three different main compliant mechanism platforms with flexure hinges consisting of a rectangular notch shape. The obvious focus of the paper is on the control of the stage with the claim of developing a model, while the design aspects and the functional principle are only briefly described and thus are not satisfactory. The content could be of relevance but this cannot be estimated to the end.

At the first impression the paper seems to be well prepared. But while reading in detail too much small and especially big questions arise which must be answered and explained through a major revision before the paper can be accepted for publication in the journal. For example, the working principle of the whole stage and the three platforms, the reason for the choice of the rectangular hinge shape and its geometric parameters, the typical hinge stroke/rotation angle of each hinge, and the influence of geometrical non-linearities and their possible consideration. Thus, most important is, that there is no clear relation of the paper’s content/focus to the journal “Micromachines” and why there is no specification of the absolute positioning value/range or the motion ranges of the three stages and the overall micro-positioner. Furthermore, the question arises why so much effort is spent to the control and why there is so less details/content about the mechanical part of the stage. In addition, there are several formal layout mistakes, to much Typos, grammar things, and the English spelling is poor, which all is not acceptable for a high-ranked scientific sound journal.

MAJOR comments:

1.) Please check English writing and avoid typos or questionable matters (see attachment).

2.) Please revise the formal layout things, e.g. paragraphs should be separated either by indentation or a plank line, often missing spaces before brackets or between values and units, symbols must be set in italics with an equation editor in the whole paper (e.g. Section 3.1), lots of symbols are not in the notation table, text fragments are not allowed and equations belongs sometimes to the sentence with required end point sign or sometimes not.

3.) Please explain more in detail the working principle as well as the loads and motion ranges of the whole stage, of each platform in the text and supported by additional figures in which the initial undeformed and the deformed state are mentioned (either drawn to scale or not drawn to scale, cf. Fig. 1 and 2). Please clarify in this context the contribution of the presented stage compared to existing stage by a quantitative comparison. Additionally coordinate systems are necessary in every design figure.

4.) What is the amplification ratio?

5.) Why is the mechanism design chosen in this configuration and how are the parameter values chosen and what values are chosen (e.g. minimum hinge thickness t, l and alpha)?

6) Why rectangular flexure hinge contours are chosen and what is with the resulting maximum stress and stress distribution in dependence of the motion range/angular stroke of each hinge? What is with the strong contour-dependent influence of the rotational axis shift on the motion precision of the whole micro-positioning stage which is expected to be in the micrometer range (e.g. line 36-40)? Please refer in this context to some of the following example references

(a) Tseytlin, Y. M.: Notch flexure hinges: An effective theory, Review of Scientific Instruments, 73, 3363–3368, doi:10.1063/1.1499761, 2002.

(b) Zelenika, S.; Munteanu, M. G.; Bona, F. De (2009): Optimized flexural hinge shapes for microsystems and high-precision applications. In: Mechanism and Machine Theory 44 (10), pp. 1826–1839. DOI: 10.1016/j.mechmachtheory.2009.03.007.

(c) Linß, S.; Schorr, P.; Zentner, L. (2017): General design equations for the rotational stiffness, maximal angular deflection and rotational precision of various notch flexure hinges. In: Mech. Sci. 8 (1), pp. 29–49. DOI: 10.5194/ms-8-29-2017.

(d) Valentini, P. P. and Pennestrì, E.: Elasto-kinematic comparison of flexure hinges undergoing large displacement, Mechanism and Machine Theory, 110, 50–60, doi:10.1016/j.mechmachtheory.2016.12.006, 2017.

7.) What is the influence of geometrical non-linearities of the mechanical/deformation behavior on the precision and precision error of the stage?

8.) What is meant by machining error exactly? This is totally unclear. Is this related with the influence of the geometrical tolerances through manufacturing?

9.) What is meant by simulation here (cf. Fig. 6)? What are the assumptions, settings and the approach?

10.) What is meant by modeling here? What are the assumptions and the approach?

Reviewer 2 Report

This manuscript reports a positioning error model and control schemes for a 6-DOF micropositioner. Overall, the text is well organized and the approach well described. However, there are some specific issues that have to be addressed:

In the introduction (line 31) the acronym MEMS should be explained as microelectronic mechanical systems or micro-electromechanical systems. In the introduction, lines 47-50 are not clear. Revision is required. The title of Section 2 is not clear. In order to improve the clarity of Figure 1 description, it would be helpful to clearly indicate in the figure the direction x, y, and z axes. How is the rotation about the y axis generated? Which of the platforms included in the compliant mechanism is responsible for it? The quantities reported in equations 1 (and in more details in the following equations 2-4) should be indicated as vectors instead of matrices at line 118. What do the quantities “f” with varying subscripts represent in definitions 3 and 4? From Figure 1 it is difficult to identify all the geometrical quantities that then appear in equation 5. Revision of the figure is thus suggested. It is not clear how it is possible to end up with equation 11 from equations 9 and 10. Additional explanation would be useful. What does ucw represent in equation 12? At lines 255-257 it is not clear how the maximum hysteresis error is evaluated. At line 271, l6 is repeated twice in the list of the most influential parameters. English revision is suggested.

Round 2

Reviewer 1 Report

The authors have amended the paper according to the reviewer’s comments. The provided responses are satisfactory and the paper can be accepted for publication after the revision of some minor necessary changes: Please check correct English writing in the whole paper again, there are still some typos, e.g. in the reference list (especially the added new references).

Reviewer 2 Report

English revision is suggested.